# Binding and Kinetic Analysis of Human Protein Phosphatase PP2A Interactions with Caspase 9 Protein and the Interfering Peptide C9h

**DOI:** 10.3390/pharmaceutics14102055

**Published:** 2022-09-27

**Authors:** Karim Dorgham, Samuel Murail, Pierre Tuffery, Eric Savier, Jeronimo Bravo, Angelita Rebollo

**Affiliations:** 1Faculty of Medicine, Sorbonne Université, Inserm, CIMI Paris, 91, bd de l’hôpital, 75013 Paris, France; 2BFA, Université Paris Cité, Inserm 1133, 75013 Paris, France; 3AP-HP, Sorbonne Université, CRSA, 75013 Paris, France; 4Instituto de Biomedicina de Valencia IBV-CSIC, Jaime Roig, 11, 46010 Valencia, Spain; 5Faculty of Pharmacy, UTCBS, Université Paris Cité, Inserm 1267, 75006 Paris, France

**Keywords:** biolayer interferometry, PP2A, Caspase 9, circular dichroism, binding affinity, C9h interfering peptide

## Abstract

The serine/threonine phosphatase PP2A and the cysteine protease Caspase 9 are two proteins involved in physiological and pathological processes, including cancer and apoptosis. We previously demonstrated the interaction between Caspase 9 and PP2A and identified the C9h peptide, corresponding to the binding site of Caspase 9 to PP2A. This interfering peptide can modulate Caspase 9/PP2A interaction leading to a strong therapeutic effect in vitro and in vivo in mouse models of tumor progression. In this manuscript, we investigate (I) the peptide binding to PP2A combining docking with molecular dynamics and (II) the secondary structure of the peptide using CD spectroscopy. Additionally, we compare the binding affinity, using biolayer interferometry, of the wild-type protein PP2A with Caspase 9 and vice versa to that observed between the PP2A protein and the interfering peptide C9h. This result strongly encourages the use of peptides as new therapeutics against cancer, as shown for the C9h peptide already in clinical trial.

## 1. Introduction

Targeting protein/protein interactions are considered a good therapeutic strategy in several pathologies. Indeed, interfering peptides blocking a specific protein/protein interaction are considered important tools to manipulate a given interaction. Moreover, administration routes, stability, pharmacokinetic parameters, and safety of the peptides have made these molecules an attractive new generation of drugs [1]. Several interfering peptides have been identified and validated in vitro and in vivo [2,3,4,5] as new potential medicaments, and some of them are in clinical development or already in the market (PEP 010, clinical trial, NCT04733027, [6,7]).

Protein phosphatase 2A (PP2A), a serine/threonine phosphatase, has been shown to have a pro-apoptotic, and also in some cases, an anti-apoptotic function [8,9]. PP2A is deregulated in many types of cancers and a number of other human diseases, including Alzheimer and cardiovascular diseases [10]. PP2A has been shown to be genetically altered and functionally inactivated in many solid cancers and leukemias. Inhibition of PP2A activity is critical to promote cell transformation, tumor progression, and angiogenesis, which indicates that PP2A has a tumor suppressive role [11,12,13]. Recent reports show that pharmacological restoration of PP2A tumor-suppressor activity effectively antagonizes cancer development and progression.

Several partners of the PP2A have been identified, including the cysteine protease Caspase 9 [14]. Caspase 9 is the initiator of intrinsic apoptosis regulating physiological cell death and pathological tissue degeneration. Clinical reports suggest that alterations in Caspase 9 expression or activation can be involved in several diseases, including cancer [15].

We previously described the interaction between the Caspase 9 and PP2A and identified the binding peptide, also called interfering peptide, able to block this interaction [14]. This peptide (C9h) derived from Caspase 9 sequence, which includes residues involved in binding to PP2A, has been validated in vitro and shows an anti-tumoral effect in vivo on xenograft models of breast cancer [14].

In this manuscript, we analyze and compare the affinity of PP2A to its protein partner, Caspase 9 and to C9h, the interfering peptide from Caspase 9 sequence, as well as the molecular dynamics and peptide structure.

## 2. Materials and Methods

### 2.1. Peptide Synthesis and Sequence

The C9h peptide was synthesized in an automated multiple peptide synthesizer with solid-phase procedure and standard Fmoc chemistry by GL Biochem (Shanghai, China). The purity and composition of the peptide were confirmed by reverse phase high-performance liquid chromatography (HPLC) and by mass spectrometry (MS). The sequence of the peptide, isolated from Caspase 9 protein, is:

Y V E T L D D I F E Q W A H S E D L

### 2.2. Docking

To generate initial poses, we have used PEP-FOLD using the sOPEP2 force-field [16], taking as constraints positions generated from the amino acids of PP2A identified by PEPscan as being at the interface between catalytic subunits alpha of the PP2A (PP2Acα and Caspase 9) similarly to the protocol previously described [17]. One hundred models were generated and clustered using ligand RMSD as criterion. After the initial peptide generation, a Monte-Carlo refinement was performed using 30,000 steps at 350 K. Since previous studies have shown that PEPscan most overlapping fragments [18,19] usually better correspond to the binding interface, and to accelerate calculations as well as conformational sampling during simulations, only the central part of the peptide, corresponding to sequence LDDIFEQWAH, has been considered.

### 2.3. Molecular Dynamics (MD) Simulation

Structure preparation and protonation were conducted using the pdbfixer module of OpenMM package. The complexes were solvated in truncated octahedron boxes with a padding of 1.0 nm and the TIP3P water model, as the Amber14SB [20] forcefield was used to model the protein atoms; 6 and 11 sodium ions were added to counter the charge of the C9h in aqueous solvent and C9h-PP2A complex systems, respectively. Simulations were computed using the OpenMM 7.6 package [21]. Systems were minimized up to 1000 steps and equilibrated for 10 ns in NPT ensemble, and temperature and pressure were equilibrated using a Monte Carlo barostat at a temperature of 300 K and a 1 atmosphere pressure. We used a 4 fs integration time step using heavy hydrogen assigned a mass of 3 atomic mass units.

Simulations were performed using the OpenMM library, with the original OpenMM script for simulated tempering (ST) simulations [22] written by Peter Eastman, modified to implement the weight calculation of Park and Pande [23] and an on-the-fly weight calculation developed by Nguyen et al. [24]. During simulated tempering (ST) simulations, 21 and 32 temperature ladders were chosen spaced exponentially between 300 and 400 K, for C9h in aqueous solvent and C9h-PP2A complex systems, respectively. Temperature exchanges were attempted every 2 ps. During C9h-PP2A complex ST simulation, and to avoid high deformation of the protein during ST, position restraints were applied on the PP2A Cα atoms of 10.0 KJ/mol/nm^2^, while the Caspase 9 fragment was free. However, to prevent the peptide from moving too far away from the initial position, some weak distance restraints were applied on the peptide backbone atoms of 100.0 KJ/mol/nm^2^, if the peptide backbone centroid moves more than 15 Å away from its initial position. Two PEPFOLD poses were simulated with ST during 4.4 and 4.6 μs. MD simulations analysis were conducted using the MDAnalysis python library [25]. For simulation in isolation, the complete C9h peptide was considered.

### 2.4. Circular Dichroism Analysis

Wild-type Caspase-9-derived synthetic peptide covering PP2Acα binding site (corresponding to residues 363–380) was synthesized. Circular dichroism was performed on a Jasco J-010 spectropolarimeter to study the presence of α-helix.

The peptide was dissolved in 10mM sodium phosphate pH 7.5 prepared from 10 mL solution A + 84 mL solution B in a final volume of 200 mL.

Solution A: 2.76 g NaH2PO4·3H2O. (0.0552 g/20 mL) (AppliChem, Darmstadt, Germany).

Solution B: 5.365 g Na2HPO4·7H2O. (0.5365 g/100 mL) (Sigma, Darmstardt, Germany).

Final pH of the mix was adjusted to 7.65.

To induce α-helix formation, a titration with trifluoroethanol (TFE) (Sigma) was performed. Samples were prepared with 50 μM of peptide dissolved in a final volume of 300 μL with increasing concentrations of TFE (20%, 40%, 60%, and 80% *v*/*v*). Samples were introduced in the spectropolarimeter in a Quartz Suprasil Precision cell 0.1 cm cuvette (Hellma), using 300 μL of buffer solution as blank. Measurements were repeated 10 times at 20 °C for each sample and 5 times for each blank. Data were processed with spectropolarimeter software to subtract blank from sample spectra and millidegrees units were further converted to molar ellipticity.

CD spectrum predictions from the MD simulations were performed using the PDBMD2CD [26] web server. Conversion from delta ellipticity (DE) to molar ellipticity (ME) was performed using the rule: ME = 3298 × DE.

### 2.5. Characterization of PP2A and C9h Peptide Interaction by ELISA

A total of 100 μL of biotinylated peptides diluted at 100 μM in PBS was incubated for 2 h at room temperature in a 96-well Streptavidine-coated plate (Pierce, Illkirch, Strasbourg, France). Wells were washed five times with PBS/0.05% Tween-20 (PBST) and filled with 100 μL of PP2A (Sigma) diluted in PBS/2.5% BSA (Sigma) at the indicated dilutions. The plate was incubated over night at 4 °C and washed five times with PBST. A total of 100 μL of rabbit polyclonal IgG anti-human PP2Aα (Santa Cruz Biotechnology, Heidelberg, Germany) was added at 5 μg/mL in PBS/BSA for 1 h at room temperature. Wells were washed 5 times with PBST and filled with 100 μL of HRP conjugated anti-rabbit IgG (Sigma) diluted at 1:20,000 in PBS/BSA for 1 h at room temperature. Wells were washed 5 times with PBST, and 100 μL of TMB substrate (Pierce) were added and incubated for 15–45 min. The reaction was stopped with 50 μL of 2 N sulphuric acid, and the absorbance was measured at 450 nm on a Multiskan EX plate reader (Thermo Scientific, Illkirch, Strasbourg, France). The Caspase 9 peptide C9h was labelled with biotin at the N-terminal and C-terminal end of the peptide.

### 2.6. Biolayer Interferometry and Kinetic Analysis

Kinetic analysis was performed based on biolayer interferometry (BLI) by using a BLItz instrument (ForteBio, Bayonne, France) at room temperature. Prior to use, each biosensor was hydrated in sample diluent (SD: PBS1X, pH 7.0, 0.02% Tween 20, 0.1% BSA) for at least 10 min. Kinetic measurements of PP2A/Caspase 9 interactions were run with Protein A biosensors (Fortebio) and consisted of seven steps, all performed in reaction buffer, as follows: (i) initial baseline in 300 µL SD for 30 s was measured; (ii) 4 µL of mouse anti-Caspase 9 or rabbit anti-PP2A were immobilized on the sensor for 150 s at 133 nM; (iii) the immunosensor was washed with 300 µL SD for 120 s; (iv) 4 µL of purified ligand Caspase 9 at 566 nM (Origen) or PP2A at 694 nM (LSBio) were loaded on the immunosensor for 150 s; (v) the ligand-coated immunosensor was washed with 300 µL SD for 120 s; (vi) association of purified analytes diluted in SD at the mentioned concentrations was studied for 150 s in 4 µL; and (vii) dissociation step was then monitored in 300 µL SD for 120 s.

Kinetic measurements of PP2A against biotinylated C9h-Cter peptide were run with Streptavidin biosensors (Fortebio, #18-5019). The initial step consisted of coating the biosensor with the peptide diluted in SD at 100 µM for 180 s in 4 µL (not shown) and was followed by three steps for the kinetic study: (i) initial baseline in 300 µL SD for 30 s was measured; (ii) association of PP2A at the indicated concentrations for 180 s in 4 µL was used; and (iii) dissociation step was then monitored in 300 µL SD for 150 s.

Experiments using empty immunosensors (no ligand loaded) were run as control. Sensorgrams were fit globally to a 1:1 binding model by BLItz Pro software, from which the association (kon) and dissociation (koff) rate constants and apparent affinities (KD) were calculated. X2 is the sum of squared deviations between the actual data point and the fitted curve. Values close to zero indicate a good curve fit. The R2 coefficient of determination is a statistical measure of how well the fitted curve approximate the real data points. Values of R2 near 1 indicate a near-perfect fit.

## 3. Results

### 3.1. Dynamic Behavior of C9h Using ST Simulations

We first present the results of ST simulations of C9h in aqueous solvent for a 9.5 microseconds simulation. As can be seen from Figure 1A, the energy landscape at 300 K is rather flat, the barriers between the different conformations being on the order of less than 3 to 4 kJ/mol. One also observes a large variability in the conformations of the representative conformations (depicted for each of the 11 clusters identified using k-mean clustering algorithm). Over all 300 K frames, the average helical content is of 36%. The most populated one (cluster 5–13%) shows propensity for helical conformations at the N- and C-terminus, whereas less populated ones (cluster 8–8% or cluster 6–7%) adopt alpha-helical conformations in the middle of C9h. Other ones (such as cluster 4–9%) adopt a largely unstructured conformation. Over all 300 K MD frames, as shown Figure 1B, the central part of the peptide shows the largest propensity to adopt helical conformations but for no more than 60% of the frames, meaning that coiled conformations occur at frequencies of more than 40% over all positions—no beta conformation being observed. It has to be noted that the force field used for the simulations (Amber14SB [20]) is not designed to simulate disordered peptides and may favor helical structures. Overall, our simulation suggests that C9h is rather flexible with a strong tendency toward helix.

### 3.2. Peptide Secondary Structure Evaluation by Circular Dichroism (CD)

The circular dichroism (CD) has been exploited for protein and peptide folding, conformational changes, intramolecular interactions, and ligand binding studies. We were interested in analyzing whether the interfering peptide (IP) C9h, a fragment of the human Caspase 9 involved in binding to the phosphatase PP2A, can maintain the secondary structure shown in the context of Caspase 9 protein. To this end, a CD analysis of the peptide was performed. Figure 2 shows that a concentration of 50 μM C9h peptide behaves as a random coil in 10 mM of sodium phosphate pH 7.5. In the presence of increasing amounts of trifluoroethanol (TFE), the structure pattern was progressively adopting a higher percentage of alpha helical structure as indicated by circular dichroism spectroscopy. The percentage of helix deduced from the CD spectra are of 2.62, 7.72, 21.58, 24.34, and 31.01% for TFE concentrations of 0, 20, 40, 60, and 80% *v*/*v*, respectively.

It is possible to compare the experimental spectrum with that inferred from the MD simulations (Figure 3A), although the correspondence between the experimental and predicted CD spectra must be taken with caution. For 208 nm, the average molar ellipticity of the predicted CD spectrum value is to −15028, i.e., clearly showing a propensity toward helix.

### 3.3. Molecular Modeling of Peptide Interaction with PP2A

PEP-FOLD poses best target (i) the active site, (ii) a position between two helices at positions 26–40 and 141–151 of PP2Aca, and (iii) a position on top of the loop encompassing residues 175–193. The two latter positions in contact with the loop 175–193 have been simulated using simulated tempering for 4.39 and 4.58 μs, respectively.

To obtain a better sampling of peptide position during MD simulations, we used the simulated tempering method [22] starting from two different PEP-FOLD poses. During ST simulation, starting on top of the loop encompassing residues 175–193, the peptide structure moves rapidly after 300 ns and finds a stable position until the end of the simulation (for 4.2 μs). This position will be referred to as cluster 1 (Figure 4A). The RMSD of peptide backbone atoms at 300 K frames, relative to the central structure of the cluster 1 after 300 ns, was 3.3 ± 1.2 Å. The low RMSD denotes a stable position of the peptide. To be noted, the peptide was mainly structured in alpha helix, with the exception of the three last residues. Concerning ST simulation starting from a position between two helices at positions 26–40, the peptide position deviates slightly more, and a stable position could be identified between 1.4 and 3.8 μs (for 2.4 μs). This position will be referred as cluster 2 (Figure 4A). The RMSD of peptide backbone atoms at 300 K frames, relative to the central structure of the cluster 2 between 1.4 and 3.8 μs, was 4.3 ± 2.1 Å. The peptide was also mainly structured as an alpha helix, with the exception of the three first residues. The two ST simulations were aggregated and submitted to a PCA analysis (Figure 4B), and frames from both simulations were then clustered using the HDBSCAN algorithm. HDBSCAN identifies 11 clusters with 7.9% of frames not part of any cluster. The two main clusters were cluster 1 (35.46% of all frames) and cluster 2 (26.93%), as the third biggest cluster concerned only 8.4% of frames. For all clusters, the average structure of the peptide was computed, the closest frame to this average structure was considered as the cluster reference structure. Reference structures of clusters 1 and 2 are represented in Figure 4A. In summary, the simulations strongly suggest the existence of stable poses of the peptide, compatible with rather low affinity values. Both best poses are in the vicinity of the PP2A loop identified by PEPscan as interacting with Caspase 9. However, even using advanced sampling techniques, a unique binding site does not seem to emerge.

Finally, it is also interesting to compare the behavior of the peptide in isolation and in interaction with PP2A. As shown Figure 3B, the CD spectra predicted for clusters 1 and 2 clearly show that for cluster 2 the spectrum is similar to that of C9h in the solvent whereas the helical signal is strengthened for cluster 1. This suggests that the cluster 1 pose might have a higher entropic cost upon binding to PP2A than the cluster 2 pose, despite it being the pose that contacts directly the PP2A loop identified by PEPscan. The possible impact on the binding affinities of the two poses remains, however, largely speculative at this point.

### 3.4. Qualitative Interaction between PP2A and the Biotinylated C9h Peptide

Biotinylation of peptides is an efficient method to specifically bind peptides to streptavidin-coated surfaces. Nevertheless, the positioning of the biotin tag in a peptide sequence can markedly influence binding interaction. In this regard, we tested whether the biotin label should be added on the N- or C-terminus of the C9h peptide. To this end, we tested by ELISA the binding of PP2A protein to the biotinylated C9h-Nter and C9h-Cter peptides, immobilized on streptavidin-coated plate. Figure 5 shows that the biotinylated peptides C9h-Cter and C9h-Nter peptides bind to the PP2A protein in a specific manner. However, the biotinylated C9h-Cter peptide shows stronger signal recognition than that of C9h-Nter. An irrelevant biotinylated peptide was used as a negative control. According to this result, we used for further experiments the biotinylated C9h-Cter peptide.

### 3.5. Affinity of PP2A Protein to Caspase 9 Protein and C9h Peptide

Kinetic parameters of protein/protein (Caspase 9/PP2A or vice versa) or protein/peptide (PP2A/C9h) interaction were measured by biolayer interferometry (BLI). BLI is an optical characterization method used to monitor interactions between label-free molecules in real time. It is based on the wavelength shift reflecting the change in thickness of the biological layer caused by the binding of molecules to the probe. After ensuring the absence of unspecific binding of Caspase 9 and PP2A to the protein-A biosensor, specific antibodies for Caspase 9 and PP2A were immobilized onto the probes. After a washing step, Caspase 9 and PP2A were respectively loaded onto the probes to constitute the working biosensors of interest. First, the relative PP2A interaction to the immobilized Caspase 9 molecule was studied (Figure 6A). Affinity measurements were conducted with varying concentrations of PP2A to determine the range of equilibrium binding constant values (KD = 8.94 × 10^−8^ M). In a vice versa procedure, PP2A was immobilized onto the probe (Figure 6B) and different concentrations of Caspase 9 protein were used to determine the constant at equilibrium (KD = 2.28 × 10^−7^ M). Overall, in both models, PP2A and Caspase 9 proteins were observed to bind with relatively good affinity (close to 10^−7^ M). Figure 6C summarize the results of constants of association, dissociation, and affinity.

Kinetic measurements of PP2A against biotinylated C9h-Cter peptide were run by using streptavidin biosensors (Figure 6D). Different concentrations of PP2A protein were loaded onto the immobilized C9h-Cter peptide to determine the constant at equilibrium (KD = 7.8 × 10^−7^ M). Unspecific binding of PP2A to the sensor tip in absence of peptide was checked using PP2A at the same working concentrations. It is of note that the abrupt increase in signal looks like a bump artifact occurring when the streptavidin biosensor moves during the transition from the association to the dissociation step. In this conformation, the dissociation rate of PP2A was approximatively one log higher than those observed with the immobilized Caspase 9 protein resulting to a slightly lower affinity. Figure 6E summarize the values of constants of association, dissociation, and affinity.

In conclusion, these results demonstrate an effective interaction between PP2A and Caspase 9 proteins, as well as a good affinity between PP2A and the interfering peptide corresponding to the sequence of Caspase 9 involved in binding to PP2A. Therefore, C9h peptide can be used as a tool to manipulate this interaction.

## 4. Discussion

In this study, we have first investigated the behavior of C9h in isolation, using both in silico and in vitro techniques. Circular dichroism (CD) spectroscopy is a widely used technique for the study of protein/peptide structure [28]. In the past decade, several algorithms based on quantitative analysis of CD spectroscopy have been proposed to predict the secondary structure content [29,30]. In the absence of high-resolution structures, CD is regarded. The isolated C9h peptide does not seem to show its natural helical structure, which was, however, achieved easily in the presence of an alpha helical inductor, such as TFE. This result suggests that, despite being isolated from its natural environment within Caspase, C9h still has the ability to maintain its helical character, which may be important to exerting biological and biochemical actions. MD simulations suggest for their part a much stronger helical content although associated with a large conformational diversity. The direct comparison between the two approaches is, however, complex, because the impact of the force field used for simulations is not well quantified, and CD prediction from structure also comes with uncertainty. These predictions have, however, interest as a means to compare the conformation ensembles of the peptide in isolation or interacting with PP2A.

In a second part, we have turned to the interaction of the peptide with PP2A. Two poses compatible with low affinity values were identified. Compared to classical MD, the ST approach we have used has improved sampling ability, but no convergence of the two MD simulations starting from two different poses could be observed. Longer simulations might be needed to conclude more strongly on the putative peptide affinities, and in parallel, using higher maximum temperature during the ST simulations could also allow a better sampling of the peptide structure. It is also possible that the peptide might interact with different patches on PP2A surface. However, the conversion of the time they bind to a patch in terms of affinity remains presently out of reach. Of note, however, both poses identified are close to the PP2A loop identified, and none really explored the region of the catalytic site of PP2A. Interestingly, also in the two proposed poses, a helical conformation is observed in different regions of the sequence suggesting that the peptide could be rather flexible. Indeed, additional clusters identified during ST simulations show intermediary non-structured forms of the peptide, which is consistent with the results of the CD experiment and suggest that the peptide could adopt a helical conformation only when binding to PP2A.

Finally, we have measured the affinity of two interacting proteins, Caspase 9 and PP2A, as well as the interaction between PP2A and the interfering peptide C9h, which has been previously identified as Caspase 9 binding site to PP2A using PEPscan approach [14], This interfering peptide was used to measure the affinity for its partner, the PP2A protein, showing a similar affinity to that obtained using the whole protein Caspase 9. To our knowledge, this is the first affinity binding constant reported between PP2A and Caspase 9.

Biolayer interferometry approach has proven effective in measuring the binding affinity of different partners, such as protein/protein or protein/peptide molecules [31,32,33]. Studying the interaction between proteins and their ligand partners using conventional methods often requires large amounts of substrates and multistep experimental methods. These concerns prevent the easy and accurate quantification of the strength of an interaction [34,35]. BLI is an optical technique for real-time measurement of macromolecular interactions. The objective is achieved through the analysis of interference with the white light, reflected by the biosensor surface. In a typical BLI experiment, the ligand is immobilized on the biosensor tip and then allowed to interact with the analyte [36]. Using BLI, a quantitative set of equilibrium binding affinities (KD) and rate of association and dissociation (Kon, Koff) can be measured in minutes using nanomolar quantities of sample. Thereby, in this work, PP2A/Caspase 9 interaction studies were conducted in the range of 173 to 694 nM and 283 to 1133 nM for PP2A and Caspase 9 proteins, respectively. The results showed that the binding affinity of PP2A to Caspase 9 is in the range of 150 nM, which can be considered as a good affinity between two protein partners. Moreover, PP2A binds to the C9h peptide with a slightly lower affinity, around 800 nM, i.e., slightly worse, which might possibly be related to C9h helical structuration upon binding.

## 5. Conclusions

In conclusion, our work suggests that the interfering peptide C9h, isolated from its molecular environment into the Caspase 9 protein, remains able to associate with PP2A in a manner relatively comparable to that observed when using the whole Caspase 9 protein. This is the first time that the interaction affinity between two proteins involved in tumoral transformation, PP2A and Caspase 9, has been measured. Overall, these results may explain the efficacy observed in vitro and in vivo of the use of C9h peptide to block PP2A/Caspase 9 interaction. Finally, our result opens the possibility to the use of peptides as tools to manipulate protein/protein interaction and its use of therapeutic peptides, as it is the case for C9h peptide.

## Figures and Tables

**Figure 1 pharmaceutics-14-02055-f001:**
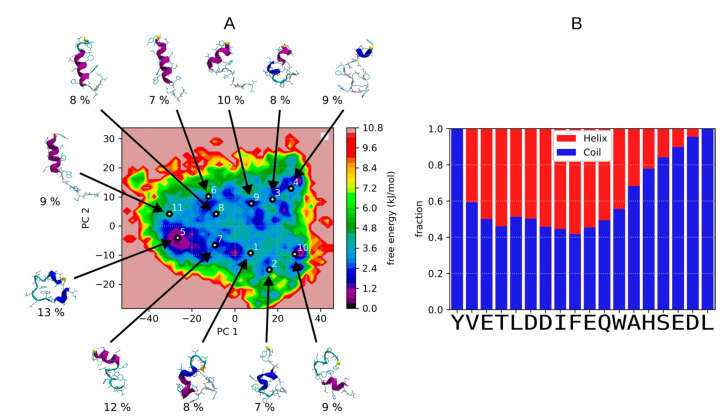
Structural landscape explored during ST simulations of C9h in aqueous solvent: (**A**) Free energy landscape at 300 K frames projected on principal component 1 (36% of variance) and 2 (10% of variance). Principal component analysis (PCA) applied to the 300 K frames ST simulation dataset of Caspase 9 peptide. Conformations representative of the clusters identified using k-means algorithm with k = 11 clusters (k has been chosen using the minimum silhouette score value for k values between 2 and 20) are depicted for each cluster together with the population of each cluster (%). (**B**) Analysis of the fraction of secondary structure adopted at each position of the peptide over all 300 K frames. Assignment of secondary structures was computed using DSSP [27].

**Figure 2 pharmaceutics-14-02055-f002:**
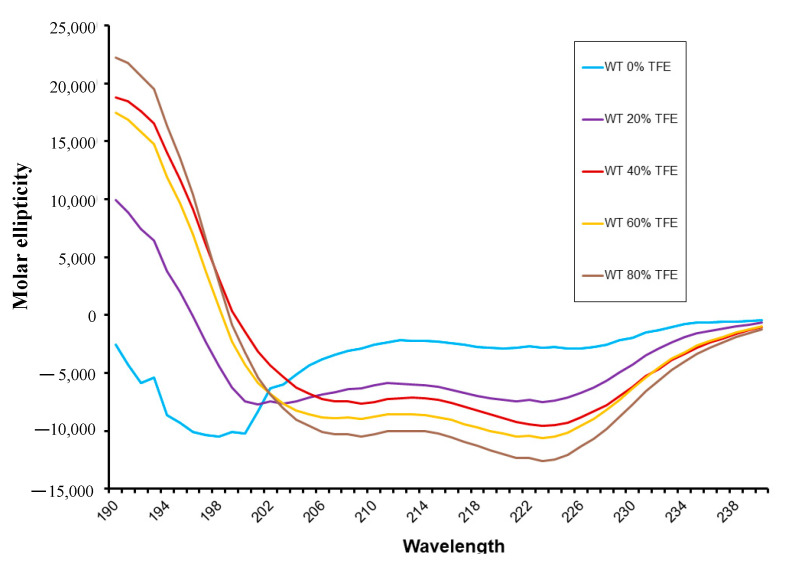
CD of C9h peptide. Data of molar ellipticity versus wavelength in circular dichroism (CD). Data obtained with peptide C9h in 10mM sodium phosphate pH 7.5 (Sky blue curve) are shown together with CD data using 20% (purple), 40% (red), 60% (orange), and 80% (brown) *v/v* of trifluoroethanol (TFE) in the same buffer. C9h shows a double CD signal at 208 and 222 nm proportional to increasing amounts of TFE indicating a-helix secondary structure content.

**Figure 3 pharmaceutics-14-02055-f003:**
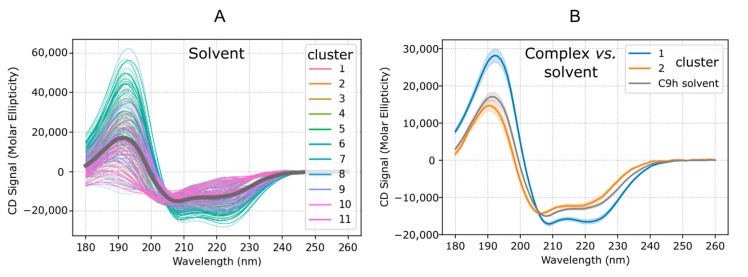
Predicted CD of C9h peptide. Data of molar ellipticity versus wavelength predicted from the MD simulation: (**A**) Spectra predicted for 702 frames periodically taken from the MD trajectory at 300 K (1 step out of 20). Color shades are attributed to the clusters as assigned in Figure 1A. The dark curve corresponds to the average over all conformations. (**B**) Average CD spectrum for C9h in solvent and simulations of C9h in interaction with PP2A starting from different positions. Clusters 1 and 2 correspond to the clusters described in Section 3.3.

**Figure 4 pharmaceutics-14-02055-f004:**
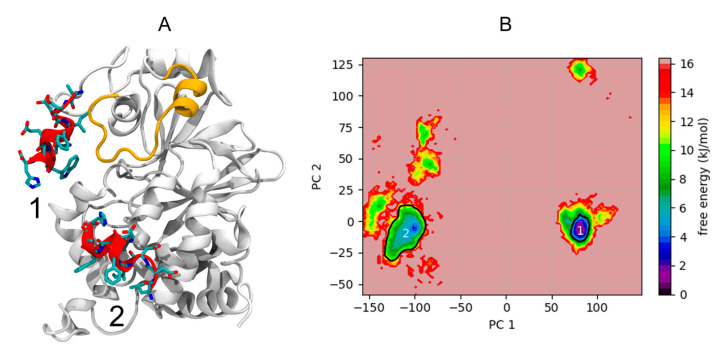
Structural landscape explored during ST simulations: (**A**) Central poses of clusters 1 and 2 are represented for the peptide as red cartoon and colored licorice, while the structure of PP2A is represented as white cartoon (only the structure of cluster 1 is represented for clarity), and the loop D172 to G190 is represented as an orange cartoon. (**B**) Free energy landscape projected on principal components 1 and 2. Principal component analysis (PCA) applied to the whole 300 K frames of ST MD simulation dataset of Caspase 9 peptide, after structural alignment on PP2A backbone atoms. Projection of first (88% of variance, *y*-axis) and second (6% of variance, *x*-axis) PCA modes computed on peptide backbone atoms. The white outline represents the contour of positions of clusters 1 and 2 frames.

**Figure 5 pharmaceutics-14-02055-f005:**
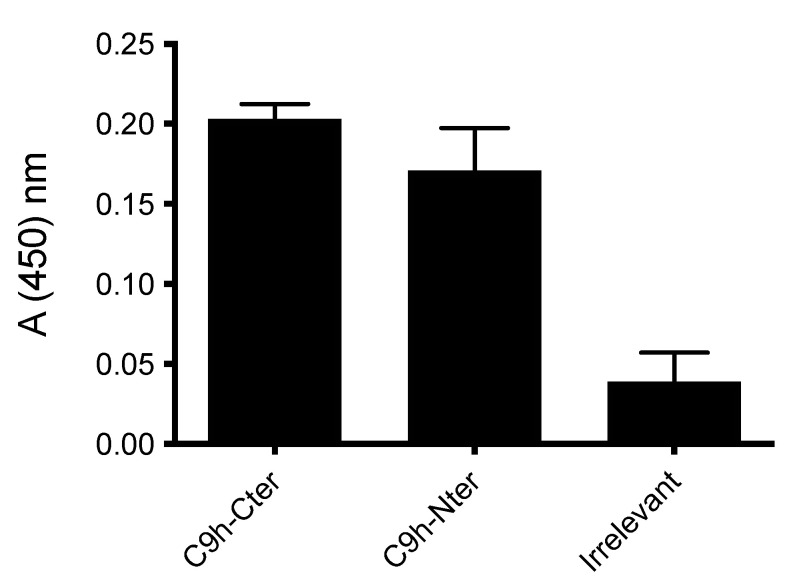
Detection of PP2A binding to C9h peptide by ELISA. Biotinylated C9h-Cter and C9h-Nter biotinylated peptides were immobilized on a Streptavidin-coated plate and incubated overnight with PP2A catalytic subunit at 0.6 μg/mL. After washing, rabbit anti-PP2A was added in each well and incubated 1 h at room temperature. Wells were washed and filled with a dilution of HRP-conjugated anti-Rabbit secondary antibody. Binding activity of PP2A is expressed as mean OD at 450 nm of duplicate wells, and bars indicate SD. These data are representative of two independent experiments.

**Figure 6 pharmaceutics-14-02055-f006:**
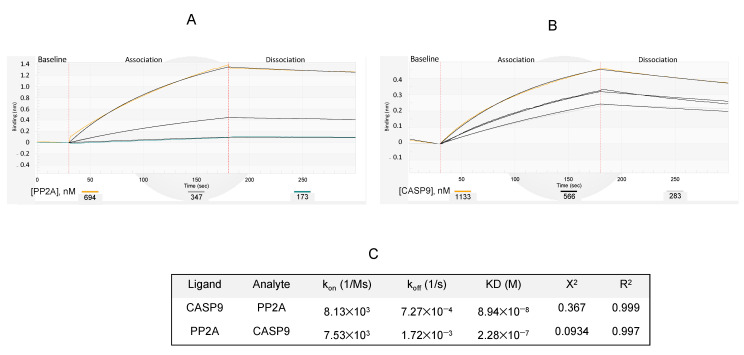
Blitz sensorgrams of PP2A/Caspase 9 interaction. The biolayer interferometry (BLI) signal is given in relative intensity units (nm): (**A**) Caspase 9 ligand was immobilized at 566 nM onto a Protein A biosensor previously coated with a mouse anti-Caspase 9 monoclonal antibody. The association and dissociation steps of the PP2A analyte injected at concentrations of 649, 347, and 173 nM were shown. (**B**) PP2A ligand was immobilized at 694 nM onto a Protein A biosensor previously coated with a rabbit anti-PP2A polyclonal antibody. Association and dissociation of the Caspase 9 analyte loaded at 694, 347, and 173 nM were shown. (**C**) Binding rate constants and apparent affinities of PP2A/Caspase 9 interactions in BLI were calculated with BLItz Pro Software. (**D**) PP2A was used at 463, 556, and 694 nM as indicated, and sample diluent (PP2A, 0 nM) was designed as sample for reference subtraction. Unspecific binding of PP2A to the sensor tip in absence of peptide was checked using PP2A at the same concentrations. (**E**) Binding rate constants and affinities of C9h-Cter peptide and PP2A were calculated with BLItz Pro Software.

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
