# Peer review of "Binding and Kinetic Analysis of Human Protein Phosphatase PP2A Interactions with Caspase 9 Protein and the Interfering Peptide C9h"

_pharmaceutics, 2022, doi:10.3390/pharmaceutics14102055_

Round 1

Reviewer 1 Report

Manuscript ID: pharmaceutics-1848639

“BINDING AND KINETIC ANALYSIS OF HUMAN PROTEIN 2 PHOSPHATASE PP2A INTERACTIONS WITH CASPASE 9 3 PROTEIN AND THE INTERFERING PEPTIDE C9H”

By: Karim Dorgham, Samuel Murail, Pierre Tuffery, Eric Savier, Jeronimo Bravo and Angelita Rebollo

The manuscript reports on investigating the interaction of the proteins serine/threonine phosphatase PP2A and cysteine protease Caspase 9 mediated by the interfering peptide C9h (which might have some pharmaceutical importance). The study is based on diverse methods encompassing Binding affinity study, MD and molecular docking, circular dichroism. The study is interesting, but I am tempted to define it a bit incomplete; it would be much more valuable, if the following points could be taken into account:

1)First, presentation: I cannot see a “fil rouge” among the different parts. For example, why is it necessary or, at least useful, that peptide C9h have random coil (or non-alfa helix) conformation in order to favor interaction between Caspase 9 and PP2A?

2)The MD calculations give the main conformers of C9h. The latter may be obtained by analyzing the CD spectra with the use of statistical analysis of spectra (as of Sreerama and Woody, Hirst, Greenfield, Fleischauer, Kardos et al, etc. (see for example  DOI: 10.1073/pnas.1500851112)). The Jasco software (possibly in the hands of the authors) might provide the authors the possibility to analyze the spectra of Figure 2, and predict the amount of alfa, beta, PP2, random coil conformers, to be compared with the predictions of MD.

3)Once the main conformers had been clustered and ranked percentagewise by MD, even DFT calculations could become possible for directly predicting the spectra: in this way a further back-up to the secondary structure of peptide C9h predicted by CD would be possible.

4)Finally, a very minor point: on page 4, line 162 there appears “method8”. Did the authors want to introduce a reference?

Reviewer 2 Report

The manuscript Dorgham et al describes some molecular characteristics of an interfering peptide which has been already published by the group. This interesting peptide disturbs the Caspase 9/PP2A protein/protein interaction and thus leading to increased apoptosis in cancer cells in vitro and in vivo and a similar peptide sequence already entered a clinical trial.

The authors first explored the possible binding site of the peptide to the PP2A via simulations. Their results showing two possible poses for the peptide. In their discussion they mentioned further optimizations for the simulations to clarify the binding site or the peptide simply has the “flexibility” to bind to the PP2A at two sites. As reader the main information I got from the simulation was the binding area (the PP2A loop) and the peptide did not block the catalytic site. These results are more or less known from the in vitro data since the peptide works. Although the authors critically discuss the results of the simulations, are real experiments (e.g. crystallographic studies or solid state NMR) combined with improved simulation settings a better way to study the C9h/PP2A interactions? Can the authors discuss this a bit more?

The CD spectroscopy results of the peptide gave a random coil structure in physiological buffer and the peptide adopted an alpha helical structure in the presence of TFE. Also, the simulations confirmed a helical structure of the peptide after binding to PP2A. Can the authors discuss this finding more in detail with regard to the following points: Does it mean a further chemical optimization of the peptide e.g. with non-natural amino acids leading to a helical structure without an inductor may further improve the binding affinity and finally biological activity? Or is it perfect to have initially a random coil structure in order to have more flexibility in the beginning of the interaction? And how are the results look like in the case the peptide is conjugated to the cell penetrating peptide which is normally used? Does the CPP help to initially maintain a helical structure? As this combination is likely the molecule for the application, this should be included in the paper.

The ELISA results are to know for the BLI experiments and for further modifcations but the figure looks overrepresented in the paper.

For the bio-layer interferometry experiments, the red lanes are the association step followed by the dissociation step? The single steps should me marked for better visualization of the experiment. Generally, the curves look plausible for the protein experiments but for the C9h peptide, the curves showing some alterations. There are negative results for the some assocations steps and for some runs a first increase in binding for the dissocations step? Can the authors explain this different pattern for the peptide compared to the protein measurements? And in the method section, the steps are not completely described for the peptide and the step numbers have the wrong order. Or am I wrong?

Regarding the KDs for the whole proteins, they differ about one log. Should not be the result in the same range and is the “real” KD a mean value of both? Is this an experimental limitation of this system?

In the discussion the affinity concentrations are mentioned for Caspase 9 being 150nM and for the C9h a value of 800nM. Where are these values came from, they are not the KD values from the table. And with a 5 times lower affinity of the peptide compared to the full protein, do the authors see the need or possibility for further peptide optimization to improve the affinity and thus the biological efficiency? This can be discussed also with regard for in vivo applications. Can a higher affinity leed to lower dosages needed for optimal effects?

Overall, it is a manuscript with a manageable number of methods and experiments and looks more like a short note than a full paper. The results are not overwhelming but after addressing the above mentions points in the discussion, the manuscript can be published.

Some typing errors:

L19 CD add spectroscopy

L23 shown

L70 caspase 9 space missing

L80 spaces

L87 converting mistake

L117 converting mistake

118 well

121 converting mistake

180 considered

181 in figure

200 converting mistake

206 figure legend wrong font size

300 seem not seen

Round 2

Reviewer 1 Report

Manuscript: pharmaceutics-1848639

“BINDING AND KINETIC ANALYSIS OF HUMAN PROTEIN 2 PHOSPHATASE PP2A INTERACTIONS WITH CASPASE 9 3 PROTEIN AND THE INTERFERING PEPTIDE C9H”

Authors: Karim Dorgham, Samuel Murail, Pierre Tuffery, Eric Savier, Jeronimo Bravo and Angelita Rebollo

Let me say at the outset that, considering the short time note for providing my review but also and most importantly based on the wonderful work that the authors did to respond the reviewers, I am in favor of fast publication of the manuscript. Yet, I would still like to make a couple of comments:

1)I insist that evaluation of peptide/protein conformations based on deconvolution of the experimental CD spectra from the Sreerama/Woody and from even more recent algorithms by Hungarian scientists would be quite useful. I encourage the authors to read the articles (some of which I pointed out in my first review), since I have a feeling that the authors are from Medical School rather than a Chemico-Physical School. The two types of schools need to talk to each other, especially by using the methods elaborated by the other school.

2)Added reference 26 should be corrected (the journal name is missing: “Nucleic Acid Research”). Possibly, added Reference 22 has the wrong DOI number. 

Reviewer 2 Report

All my comments were adequately addressed or explained!

Author Response

see attached document
